# Regulatory Single Nucleotide Polymorphism of the Bovine *IFITM3* Gene Induces Differential Transcriptional Capacities of Hanwoo and Holstein Cattle

**DOI:** 10.3390/genes12111662

**Published:** 2021-10-21

**Authors:** Yong-Chan Kim, Min-Ju Jeong, Byung-Hoon Jeong

**Affiliations:** 1Korea Zoonosis Research Institute, Jeonbuk National University, Iksan 54531, Korea; kych@jbnu.ac.kr (Y.-C.K.); minju5149@jbnu.ac.kr (M.-J.J.); 2Department of Bioactive Material Sciences and Institute for Molecular Biology and Genetics, Jeonbuk National University, Jeonju 54896, Korea

**Keywords:** cattle, IFITM3, FMDV, IAV, SNP, promoter, Nkx2-1

## Abstract

Interferon-induced transmembrane protein 3 (IFITM3), a crucial effector of the host’s innate immune system, prohibits an extensive range of viruses. Previous studies have reported that single nucleotide polymorphisms (SNPs) of the *IFITM3* gene are associated with the expression level and length of the IFITM3 protein and can impact susceptibility to infectious viruses and the severity of infection with these viruses. However, there have been no studies on polymorphisms of the bovine *IFITM3* gene. In the present study, we finely mapped the bovine *IFITM3* gene and annotated the identified polymorphisms. We investigated polymorphisms of the bovine *IFITM3* gene in 108 Hanwoo and 113 Holstein cattle using direct sequencing and analyzed genotype, allele, and haplotype frequencies and linkage disequilibrium (LD) between the *IFITM3* genes of the two cattle breeds. In addition, we analyzed transcription factor-binding sites and transcriptional capacity using PROMO and luciferase assays, respectively. Furthermore, we analyzed the effect of a nonsynonymous SNP of the *IFITM3* gene using PolyPhen-2, PANTHER, and PROVEAN. We identified 23 polymorphisms in the bovine *IFITM3* gene and found significantly different genotype, allele, and haplotype frequency distributions and LD scores between polymorphisms of the bovine *IFITM3* gene in Hanwoo and Holstein cattle. In addition, the ability to bind the transcription factor Nkx2-1 and transcriptional capacities were significantly different depending on the c.-193T > C allele. Furthermore, nonsynonymous SNP (F121L) was predicted to be benign. To the best of our knowledge, this is the first genetic study of bovine *IFITM3* polymorphisms.

## 1. Introduction

Interferon-induced transmembrane protein 3 (IFITM3), a downstream effector of the interferon signal pathway in the host innate immune system, plays a protective role against several kinds of infectious viruses, including influenza A viruses (IAVs), Ebola virus (EBOV), Marburg virus (MARV), severe acute respiratory syndrome coronavirus (SARS-CoV), dengue virus (DEV), West Nile virus (WNV), Zika virus (ZIKV), foot-and-mouth disease virus (FMDV), African swine fever virus (ASFV), and SARS-CoV-2 [1,2,3,4,5,6]. Although the length and topology of the IFITM3 protein differ slightly among species, the antiviral capacity of the IFITM3 protein is related to the CD225 domain, which is well conserved between species and significantly correlated with the expression level and integrity of the IFITM3 protein [7,8,9,10,11].

In a previous study, a splicing variant inducing the single nucleotide polymorphism (SNP) rs12252 was shown to be associated with the severity of pandemic influenza A 2009 virus infection. The C allele of the rs12252 SNP was suggested to generate a truncated isoform of the IFITM3 protein (Δ21 IFITM3) and reduce antiviral capacity [3,12]. In addition, the rs34481144 SNP, which is located on exon 1 of the *IFITM3* gene, was found to be related to a mechanism by which the transcription of the *IFITM3* gene is downregulated [13,14]. An allele inducing transcriptional variation was shown to cause the severity of pandemic influenza A 2009 virus infection. Furthermore, rs6598045 SNP, which is located on the proximal promoter region of the *IFITM3* gene, is correlated with a mechanism by which transcription of the *IFITM3* gene is regulated [1]. Susceptibility to pandemic influenza A 2009 virus infection was significantly increased with the T allele of the SNP rs6598045, and a difference in transcriptional capacity was also detected. In chicken, the c.298C > A (L100M) SNP was found to induce a variation in the topology of the IFITM3 protein and increased the length of transmembrane domain 2 (TM2) [2]. In addition, the *IFITM3* gene is expressed in primordial germ cells (PGCs), and IFITM3 protein plays a pivotal role in germline development via PGCs localization [15]. However, although several polymorphisms of the *IFITM3* gene in various species are strongly associated with antiviral capacity and genetic features of the IFITM3 protein, polymorphisms of the bovine *IFITM3* gene have not been investigated thus far.

In the present study, we investigated the bovine *IFITM3* gene in Hanwoo and Holstein cattle by using direct sequencing and compared the genotype and allele frequencies of the *IFITM3* gene in the two cattle breeds. In addition, we analyzed linkage disequilibrium (LD) and haplotype frequency of polymorphisms of the bovine *IFITM3* gene. We also assessed differences in a transcription factor-binding site and transcriptional capacity based on an allele of the *IFITM3* gene with a regulatory SNP using PROMO and luciferase assays, respectively [16]. Furthermore, we annotated the effect of a nonsynonymous SNP of the *IFITM3* gene using PolyPhen-2, PANTHER, and PROVEAN [17,18,19].

## 2. Materials and Methods

### 2.1. Ethical Statement

Tissue samples from 221 cattle of 2 breeds (108 Hanwoo and 113 Holstein cattle) were provided from slaughterhouses in the Republic of Korea. All experimental procedures were approved by the Institute of Animal Care and Use Committee of Chonbuk National University (CBNU 2018-079).

### 2.2. Genetic Analysis of the IFITM3 Gene

Genomic DNA was extracted from 20 mg of brain tissue sample using a HiYield genomic DNA mini kit (Real Biotech Corporation, Banqiao Taiwan). Polymerase chain reaction (PCR) was carried out to amplify the bovine *IFITM3* gene using BioFACT™ Taq DNA Polymerase (BioFACT, Daejeon, Korea). Information on bovine *IFITM3* gene-specific primers and experimental conditions is provided in Table 1. The PCR mixture contained 2.5 µL of 10× *Taq* DNA polymerase reaction buffer, 1 µL of genomic DNA, 10 pmol of each primer, 0.5 µL of a 0.2 µM dNTP mixture, 0.2 µL of 0.04 units of *Taq* DNA polymerase, and sterile deionized water in a total volume of 25 µL. PCR amplicons were directly sequenced by using an ABI 3730 sequencer (ABI, Foster City, California, USA), and sequencing electropherograms were visualized by using Finch TV software (Geospiza, Inc., Seattle, WA, USA).

### 2.3. In Silico Analysis

PROMO was utilized to analyze transcription factor-binding sites. Two major haplotypes based on alleles containing regulatory SNPs in the proximal promoter sequences of the *IFITM3* gene were inputted and analyzed. The effect of the polymorphisms of the bovine *IFITM3* gene was evaluated by PolyPhen-2 (http://genetics.bwh.harvard.edu/pph2/ (accessed on 3 March 2021)), PANTHER (http://www.pantherdb.org/ (accessed on 3 March 2021)), PROVEAN (http://provean.jcvi.org/index.php (accessed on 3 March 2021)), and AMYCO (http://bioinf.uab.es/amycov04/ (accessed on 3 March 2021)).

### 2.4. Cell Culture

Embryonic bovine tracheal (EBtr) cells were provided by the Korea Cell Line Bank and maintained in Eagle’s minimum essential medium (ATCC, Manassas, VA, USA). In order to prepare the complete growth medium, 10% (*v/v*) fetal bovine serum (Gibco, Gaithersburg, MD, USA) was added. EBtr cells were cultured at 37°C in a humidified atmosphere of 5% CO_2_ (*v/v*) in air.

### 2.5. Plasmids and Luciferase Assay

The promoter sequences based on alleles of the *IFITM3* gene were synthesized and inserted into the pGL4.10 [luc2] vector (Promega, Fitchburg, WI, USA). Plasmid construction and preparation followed standard protocols. The plasmids were transfected using Lipofectamine (Invitrogen, Carlsbad, CA, USA) according to the manufacturer’s instructions. The transfected cells were incubated for 30 h, and the promoter activity of the *IFITM3* gene was measured with a Glomax 20/20 luminometer (Promega, Fitchburg, WI, USA) using a luciferase assay system (Promega, Fitchburg, WI, USA).

### 2.6. Statistical Analysis

Statistical analyses were performed using SAS version 9.4 (July 2013, SAS Institute, Inc., Cary, NC, USA). The differences in genotype and allele frequencies of the *IFITM3* gene between cattle breeds were compared using the χ^2^ test. The Hardy–Weinberg equilibrium (HWE) test and haplotype and LD analyses of 23 polymorphisms of the bovine *IFITM3* gene were performed using Haploview version 4.2 (September 2009, Broad Institute, Cambridge, MA, USA). Luciferase assays were carried out in three independent experiments, and statistical significance was determined by *p*-value calculated by two-tailed Student’s *t*-test for single comparisons. The symbol “***” indicates *p* < 0.001.

## 3. Results

### 3.1. Identification of Polymorphisms of the Bovine IFITM3 Gene

In order to investigate polymorphisms of the bovine *IFITM3* gene, we performed direct sequencing with *IFITM3* gene-specific primers in 108 Hanwoo and 113 Holstein cattle (Table 1). We identified a total of 23 polymorphisms of the bovine *IFITM3* gene, including 1 nonsynonymous SNP (c.361T > C, p.Phe121Leu) and 2 insertion/deletion polymorphisms (c.249+350_351delCA and c.249+395delG) (Figure 1).

### 3.2. Comparison of Genotype and Allele Frequencies of Polymorphisms of the Bovine IFITM3 Gene among Cattle Breeds

We investigated the differences in allele and genotype frequencies of the bovine *IFITM3* gene between Hanwoo and Holstein cattle. In brief, a total of 19 polymorphisms showed significantly different genotype and allele distributions between Hanwoo and Holstein cattle: c.-193T > C, c.249+36G > C, c.249+39T > G, c.249+320G > C, c.249+350_351delCA, c.249+359G > A, c.249+367G > A, c.249+372C > T, c.249+395delG, c.249+398G > C, c.249+399C > A, c.249+400A > G, c.249+401G > T, c.249+402T > G, c.249+405G > A, c.249+455G > T, c.249+472G > C, c.361T > C and c.396C > T. Among the 23 polymorphisms, the c.136C > T, c.249+350_351delCA, c.249+372C > T, c.249+395delG, c.249+455G > T, and c.249+472G > C polymorphisms were specific to Hanwoo cattle. In addition, the c.249+320G > C, c.249+64C > A, c.249+359G > A, and c.396C > T polymorphisms were specific to Holstein cattle (Table 2).

### 3.3. LD and Haplotype Analyses of the Bovine IFITM3 Gene

Since breed-specific polymorphisms were identified, we carried out LD analysis of the 23 polymorphisms of the bovine *IFITM3* gene in Hanwoo and Holstein cattle. The LD scores for the Hanwoo and Holstein cattle are shown in Table 3 and Table 4, respectively. In brief, 27 high LD scores were identified in Hanwoo cattle, including two Hanwoo-specific LD scores (between c.249+320G > C and c.249+367G > A and between c.249+395delG and c.249+472G > C). In addition, 26 high LD scores were identified in Holstein cattle, including one Holstein-specific LD score (between c.105C > G and c.249+32G > C).

We also examined the haplotype distribution of the 23 polymorphisms of the bovine *IFITM3* gene in Hanwoo and Holstein cattle. Detailed information on the haplotype distributions of bovine *IFITM3* gene polymorphisms in Hanwoo and Holstein cattle is provided in Table 5 and Table 6, respectively. In summary, a total of 12 major haplotypes were identified in Hanwoo cattle (Table 5). Among the 12 haplotypes, the TCCGCGCGWtGGCWtGCAGTGGGCC haplotype had the highest frequency (14.8%), followed by the CCCGGTCGWtGGCWtGCAGTGGGCC (11.6%) and CCCGGTCGDelGACWtCAGTGAGGTC (7.9%) haplotypes. In Holstein cattle, a total of six major haplotypes were identified (Table 6). Among these six haplotypes, the TCCGCGCGWtGGCWtGCAGTGGGCC haplotype had the highest frequency (43.4%), followed by the CCCGGTCGWtGGCWtGCAGTGGGCC (16.4%) and TCCGCGCGWtAGCWtGCAGTGGGCC (9.7%) haplotypes.

### 3.4. The Transcription Factor-Binding Capacity of the Bovine IFITM3 Gene

We found two regulatory SNPs in the proximal promoter region of the bovine *IFITM3* gene, and c.-193T > C showed significantly different genotypes and allele distributions in Hanwoo and Holstein cattle (Table 2). We analyzed the transcription factor-binding capacity of the bovine *IFITM3* gene according to the c.-193T > C alleles using PROMO. Interestingly, the haplotype with the T allele and the haplotype with the C allele differed in their ability to bind transcription factor Nkx2-1 (Figure 2).

### 3.5. Promoter Activities Based on Alleles of Regulatory SNPs in the Proximal Promoter Region of the Bovine IFITM3 Gene

We investigated the differences in promoter activity according to the alleles of promoter SNPs, which showed different genotype and allele frequencies in Hanwoo and Holstein cattle (Table 2). The T-type promoter, which contained the T allele of c.-193T > C, was more prevalent in Hanwoo cattle. The C-type promoter, which contained the C allele of c.-193T > C, was more prevalent in Holstein cattle. Notably, the C-type promoter significantly increased the expression level of mRNA compared to that of the T-type promoter (Figure 3).

### 3.6. In Silico Annotation of a Nonsynonymous SNP of the Bovine IFITM3 Gene

The impact of the nonsynonymous SNP c.361T > C (F121L) of the bovine *IFITM3* gene identified in this study was analyzed by PolyPhen-2, PANTHER, and PROVEAN. Notably, the F121L mutation was predicted to be benign by all three programs (Table 7).

## 4. Discussion

Previous studies have reported that overexpression of the IFITM3 protein inhibits an extensive range of viruses under experimental conditions [20,21,22]. This propensity has been consistently reported under natural conditions. The duck species that are known to be resistant to avian influenza virus showed elevated expressions of the IFITM3 protein compared to that in the chicken species that are known to be susceptible to avian influenza virus. In addition, the genetic polymorphisms of Ross chickens, a kind of broiler, were significantly different from those of Dekalb White chickens, a kind of layer. Broilers show more resistance to viral infection than layers, a notable feature of the *IFITM3* gene [2]. In humans, the SNPs rs34481144 and rs6598045 of the human *IFITM3* gene, which are related to the modulated expression of this gene, showed prominent associations with the severity and susceptibility of pandemic influenza A 2009 virus infection, respectively [1,13,14]. In addition, the SNP rs12252, which influences the length of the IFITM3 protein, was found to be related to susceptibility to pandemic influenza A 2009 virus infection. Furthermore, recent studies have reported that the IFITM3 protein is also involved in not only immune-related functions but also embryogenesis and feed efficiency [15,23].

Thus, in the present study, we investigated polymorphisms of the bovine *IFITM3* gene, which can affect the expression level or function of the IFITM3 protein. We found a total of 23 polymorphisms in the bovine *IFITM3* gene. In addition, Hanwoo and Holstein cattle showed significantly different genotype and allele frequencies and breed-specific polymorphisms (Table 2). The LD scores and haplotype frequencies of the polymorphisms also showed different distributions between the two cattle breeds (Table 3, Table 4, Table 5 and Table 6). These results indicate that the *IFITM3* genes in Hanwoo and Holstein cattle have significantly different genetic properties, including genotype, allele, and haplotype frequencies and LD scores. In addition, we annotated a regulatory SNP (c.-193T > C) and a nonsynonymous SNP (c.361T > C, F121L) of the bovine *IFITM3* gene. Strikingly, the C-type haplotype with the C allele of c.-193T > C and the T-type haplotype with the T allele of c.-193T > C differed in their ability to bind transcription factor Nkx2-1, and the C-type haplotype exhibited elevated expression of the *IFITM3* gene compared to the T-type haplotype (Figure 3). Since the T allele is frequently observed in Holstein cattle while the C allele is frequently observed in Hanwoo cattle, this result suggests a significant difference in the expression of the bovine *IFITM3* gene between the cattle breeds. Indeed, Nkx2-1, a member of the Nkx-homeodomain factor family, is related to the regulation of organ development. In addition, Nkx2-1 is associated with several diseases, including benign hereditary chorea, choreoathetosis, congenital hypothyroidism, and neonatal respiratory distress. Furthermore, Nkx2-1 has a function in organ development, and it is involved in morphogenesis [24,25]. Since the IFITM3 protein is also related to cellular developmental processes, further investigation of the relationship between IFITM3 and Nkx2-1 is highly desirable in the future. We also annotated a nonsynonymous SNP (c.361T > C, F121L) by using in silico annotation tools. The F121L mutation was predicted to be benign (Table 7). Since several viruses have been reported to be associated with the *IFITM3* gene, future study of the characterization of the bovine *IFITM3* gene in other local cattle breeds is highly desirable.

## 5. Conclusions

In conclusion, we finely mapped the bovine *IFITM3* gene and annotated regulatory and nonsynonymous SNPs. We identified 23 polymorphisms of the bovine *IFITM3* gene and significantly different genotype, allele, and haplotype distributions and LD scores for these polymorphisms of the bovine *IFITM3* gene between Hanwoo and Holstein cattle. In addition, transcription factor-binding ability and transcriptional capacity were significantly different depending on regulatory SNP alleles. A nonsynonymous SNP (F121L) was predicted to be benign. To the best of our knowledge, this is the first genetic report of bovine *IFITM3* polymorphisms.

## Figures and Tables

**Figure 1 genes-12-01662-f001:**
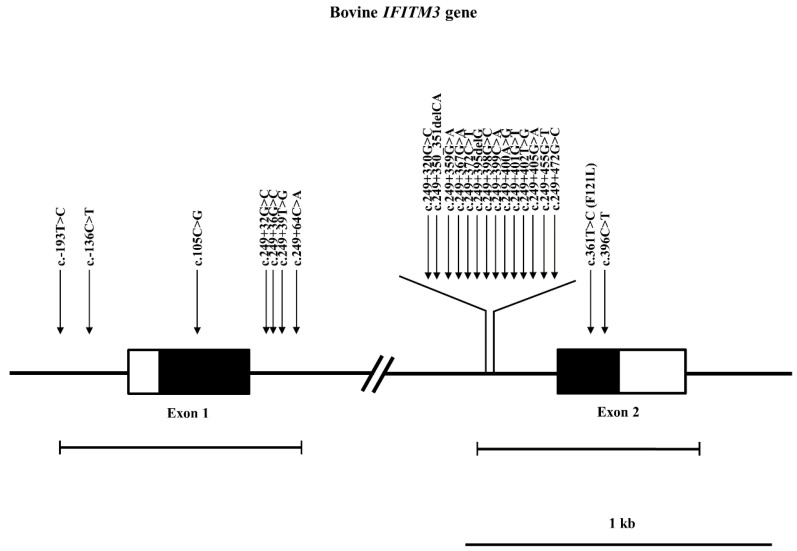
Gene map and polymorphisms of the bovine interferon-induced transmembrane protein 3 (*IFITM3*) gene on chromosome 29. The open reading frames (ORFs) in exon 1 and exon 2 are marked with black blocks, and white blocks represent the 5′ and 3′ untranslated regions (UTRs). The outlined horizontal bars indicate the sequenced regions. The 23 novel polymorphisms found in this study are indicated by arrows above the gene.

**Figure 2 genes-12-01662-f002:**
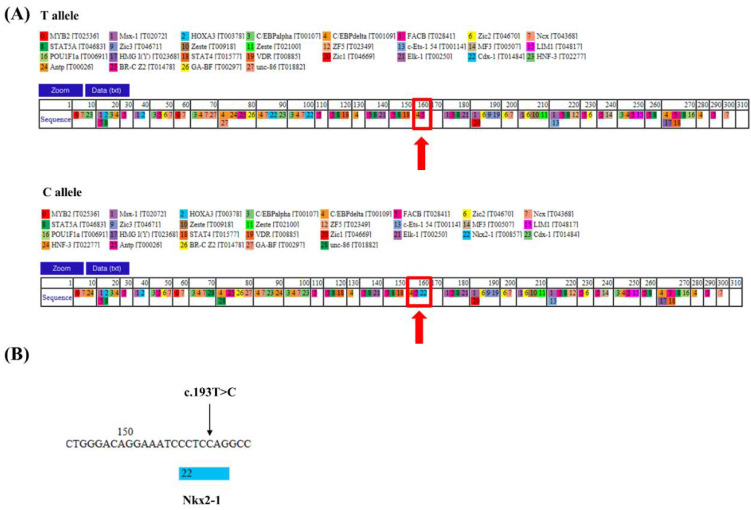
Analysis of the transcription factor-binding abilities of 2 haplotypes of the proximal promoter sequence of the bovine *IFITM3* gene. (**A**) The transcription factor-binding site according to c.-193T > C allele. Red boxes and arrows indicate differences in the transcription-binding sites of the haplotype of -193T > C with the T allele and the haplotype of -193T > C with the C allele. (**B**) Magnified view of the locus containing c.-193T > C region showing differences in Nkx2-1 binding between the haplotype with the T allele of -193T > C and the haplotype with the C allele of -193T > C.

**Figure 3 genes-12-01662-f003:**
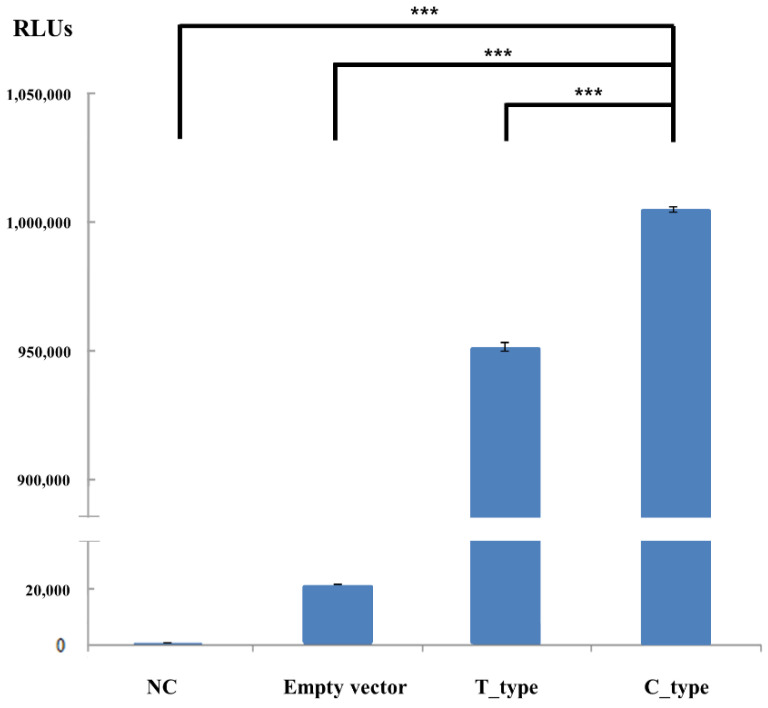
Promoter activity of the bovine *IFITM3* gene. Promoter activities of the bovine *IFITM3* gene with 2 promoter types. The symbol “***”, indicates *p* < 0.001. RLUs indicate relative luciferase light units. NC indicates negative control.

**Table 1 genes-12-01662-t001:** Detailed information on specific primer sets used for polymerase chain reaction (PCR).

Name		Size	Annealing Temperature
F1_F	GGCATTTAACGGGTGGATTCAG	774 bp	62 °C
F1_R	CATGCAGCAGAACAACACACA		
F2_F	GCCAGAGAAAGGATGGGAGA	692 bp	61 °C
F2_R	TGAAGGACAGTGACGAGAGG		

**Table 2 genes-12-01662-t002:** Genotype and allele frequencies of bovine *IFITM3* gene polymorphisms in two cattle breeds.

Polymorphism	Breed	Genotype Frequency, *n* (%)			*p*-Value	Allele Frequency, *n* (%)		*p*-Value	HWE
c.-193C > T		CC	CT	TT		C	T		
	Hanwoo	72 (66.67)	34 (31.48)	2 (1.85)	<0.0001	178 (82.41)	38 (17.59)	<0.0001	0.3729
	Holstein	11 (9.73)	46 (40.71)	56 (49.56)		68 (30.09)	158 (69.91)		0.7306
c.-136C > T		CC	CT	TT		C	T		
	Hanwoo	107 (99.07)	1 (0.93)	0 (0)	0.4887	215 (99.54)	1 (0.46)	0.4887	0.9614
	Holstein	113 (100)	0 (0)	0 (0)		226 (100)	0 (0)		0
c.105C > G		CC	CG	GG		C	G		
(p.Pro35Pro)	Hanwoo	107 (99.07)	1 (0.93)	0 (0)	1.0	215 (99.54)	1 (0.46)	1.0	0.9614
	Holstein	111 (98.23)	2 (1.77)	0 (0)		224 (99.12)	2 (0.88)		0.9244
c.249+32G > C		GG	GC	CC		G	C		
	Hanwoo	108 (100)	0 (0)	0 (0)	1.0	216 (100)	0 (0)	1.0	0
	Holstein	112 (99.12)	1 (0.88)	0 (0)		225 (99.56)	1 (0.44)		0.9623
c.249+36G > C		GG	GC	CC		G	C		
	Hanwoo	41 (37.96)	50 (46.3)	17 (15.74)	<0.0001	132 (61.11)	84 (38.89)	<0.0001	0.7872
	Holstein	12 (10.62)	45 (39.82)	56 (49.56)		69 (30.53)	157 (69.47)		0.5153
c.249+39T > G		TT	TG	GG		T	G		
	Hanwoo	41 (37.96)	50 (46.3)	17 (15.74)	<0.0001	132 (61.11)	84 (38.89)	<0.0001	0.7872
	Holstein	12 (10.62)	45 (39.82)	56 (49.56)		69 (30.53)	157 (69.47)		0.5153
c.249+64C > A		CC	CA	AA		C	A		
	Hanwoo	108 (100)	0 (0)	0 (0)	1.0	216 (100)	0 (0)	1.0	0
	Holstein	112 (99.12)	1 (0.88)	0 (0)		225 (99.56)	1 (0.44)		0.9623
c.249+320G > C		GG	GC	CC		G	C		
	Hanwoo	67 (62.04)	40 (37.04)	1 (0.92)	<0.0001	174 (80.56)	42 (19.44)	<0.0001	0.0582
	Holstein	103 (91.15)	10 (8.85)	0 (0)		216 (95.58)	10 (4.42)		0.6226
c.249+350_351delCA		WT/WT	WT/DEL	DEL/DEL		WT	DEL		
	Hanwoo	77 (71.3)	31 (28.7)	0 (0)	<0.0001	185 (85.65)	31 (14.35)	<0.0001	0.0816
	Holstein	113 (100)	0 (0)	0 (0)		226 (100)	0 (0)		0
c.249+359G > A		GG	GA	AA		G	A		
	Hanwoo	108 (100)	0 (0)	0 (0)	<0.0001	216 (100)	0 (0)	<0.0001	0
	Holstein	74 (65.49)	39 (34.51)	0 (0)		187 (82.74)	39 (17.26)		0.0401
c.249+367G > A		GG	GA	AA		G	A		
	Hanwoo	22 (20.37)	85 (78.7)	1 (0.93)	<0.0001	129 (59.72)	87 (40.28)	<0.0001	*p* < 0.0001
	Holstein	82 (72.57)	31 (27.43)	0 (0)		195 (86.28)	31 (13.72)		0.091
c.249+372C > T		CC	CT	TT		C	T		
	Hanwoo	81 (75)	26 (24.07)	1 (0.93)	<0.0001	188 (87.04)	28 (12.96)	<0.0001	0.4871
	Holstein	113 (100)	0 (0)	0 (0)		226 (100)	0 (0)		0
c.249+395delG		WT/WT	WT/DEL	DEL/DEL		WT	DEL		
	Hanwoo	90 (83.33)	18 (16.67)	0 (0)	<0.0001	198 (91.67)	18 (8.33)	<0.0001	0.3448
	Holstein	113 (100)	0 (0)	0 (0)		226 (100)	0 (0)		0
c.249+398G > C		GG	GC	CC		G	C		
	Hanwoo	15 (13.89)	91 (84.26)	2 (1.85)	<0.0001	121 (56.02)	95 (43.98)	<0.0001	*p* < 0.0001
	Holstein	76 (67.26)	37 (32.74)	0 (0)		189 (83.63)	37 (16.37)		0.0374
c.249+399C > A		CC	CA	AA		C	A		
	Hanwoo	12 (11.11)	96 (88.89)	0 (0)	<0.0001	120 (55.56)	96 (44.44)	<0.0001	*p* < 0.0001
	Holstein	86 (76.11)	27 (23.89)	0 (0)		199 (88.05)	27 (11.95)		0.1492
c.249+400A > G		AA	AG	GG		A	G		
	Hanwoo	27 (25)	81 (75)	0 (0)	<0.0001	135 (62.5)	81 (37.5)	<0.0001	*p* < 0.0001
	Holstein	89 (78.76)	24 (21.24)	0 (0)		202 (89.38)	24 (10.62)		0.2066
c.249+401G > T		GG	GT	TT		G	T		
	Hanwoo	26 (24.07)	82 (75.93)	0 (0)	<0.0001	134 (62.04)	82 (37.96)	<0.0001	*p* < 0.0001
	Holstein	88 (77.88)	25 (22.12)	0 (0)		201 (88.94)	25 (11.06)		0.1861
c.249+402T > G		TT	TG	GG		T	G		
	Hanwoo	21 (19.44)	87 (80.56)	0 (0)	<0.0001	129 (59.72)	87 (40.28)	<0.0001	*p* < 0.0001
	Holstein	85 (75.22)	28 (24.78)	0 (0)		198 (87.61)	28 (12.39)		0.1328
c.249+405G > A		GG	GA	AA		G	A		
	Hanwoo	12 (11.11)	92 (85.19)	4 (3.7)	<0.0001	116 (53.7)	100 (46.3)	<0.0001	*p* < 0.0001
	Holstein	80 (70.8)	33 (29.2)	0 (0)		193 (85.4)	33 (14.6)		0.691
c.249+455G > T		GG	GT	TT		G	T		
	Hanwoo	31 (28.7)	67 (62.04)	10 (9.26)	<0.0001	129 (59.72)	87 (40.28)	<0.0001	0.0026
	Holstein	113 (100)	0 (0)	0 (0)		226 (100)	0 (0)		0
c.249+472G > C		GG	GC	CC		G	C		
	Hanwoo	83 (76.85)	25 (23.15)	0 (0)	<0.0001	191 (88.43)	25 (11.57)	<0.0001	0.1738
	Holstein	113 (100)	0 (0)	0 (0)		226 (100)	0 (0)		0
c.361T > C		TT	TC	CC		T	C		
(p.Phe121Leu)	Hanwoo	39 (36.11)	46 (42.59)	23 (21.3)	<0.0001	124 (57.41)	92 (42.59)	<0.0001	0.1799
	Holstein	0 (0)	14 (12.39)	99 (87.61)		14 (6.19)	212 (93.81)		0.4827
c.396C > T		CC	CT	TT		C	T		
(p.Ile132Ile)	Hanwoo	108 (100)	0 (0)	0 (0)	<0.01	216 (100)	0 (0)	<0.01	0
	Holstein	103 (91.15)	10 (8.85)	0 (0)		216 (95.58)	10 (4.42)		0.6226

**Table 3 genes-12-01662-t003:** Linkage disequilibrium (LD) scores among 23 polymorphisms of the bovine *IFITM3* gene in Hanwoo.

	P1	P2	P3	P4	P5	P6	P7	P8	P9	P10	P11	P12	P13	P14	P15	P16	P17	P18	P19	P20	P21	P22	P23
P1	-	-	-	-	-	-	-	-	-	-	-	-	-	-	-	-	-	-	-	-	-	-	-
P2	0.022	-	-	-	-	-	-	-	-	-	-	-	-	-	-	-	-	-	-	-	-	-	-
P3	0.001	0	-	-	-	-	-	-	-	-	-	-	-	-	-	-	-	-	-	-	-	-	-
P4	-	-	-	-	-	-	-	-	-	-	-	-	-	-	-	-	-	-	-	-	-	-	-
P5	**0.335**	0.007	0.003	-	-	-	-	-	-	-	-	-	-	-	-	-	-	-	-	-	-	-	-
P6	**0.335**	0.007	0.003	-	**1**	-	-	-	-	-	-	-	-	-	-	-	-	-	-	-	-	-	-
P7	-	-	-	-	-	-	-	-	-	-	-	-	-	-	-	-	-	-	-	-	-	-	-
P8	0.001	0.019	0.019	-	0.001	0.001	-	-	-	-	-	-	-	-	-	-	-	-	-	-	-	-	-
P9	0.036	0.001	0.001	-	0.107	0.107	-	0	-	-	-	-	-	-	-	-	-	-	-	-	-	-	-
P10	-	-	-	-	-	-	-	-	-	-	-	-	-	-	-	-	-	-	-	-	-	-	-
P11	0.072	0.007	0.007	-	0.044	0.044	-	**0.358 ***	0.156	-	-	-	-	-	-	-	-	-	-	-	-	-	-
P12	0.032	0.001	0.001	-	0.095	0.095	-	0.036	0.268	-	0.131	-	-	-	-	-	-	-	-	-	-	-	-
P13	0.019	0	0	-	0.143	0.143	-	0.022	0.015	-	0.061	0.013	-	-	-	-	-	-	-	-	-	-	-
P14	0.168	0.006	0.006	-	0.091	0.091	-	0.175	0.132	-	**0.634**	0.044	0.071	-	-	-	-	-	-	-	-	-	-
P15	0.171	0.006	0.006	-	0.081	0.081	-	0.256	0.209	-	**0.73**	0.084	0.073	**0.907**	-	-	-	-	-	-	-	-	-
P16	0.128	0.008	0.008	-	0.056	0.056	-	**0.318**	0.234	-	**0.775**	0.121	0.055	**0.651**	**0.75**	-	-	-	-	-	-	-	-
P17	0.131	0.008	0.008	-	0.058	0.058	-	0.311	0.229	-	**0.792**	0.116	0.056	**0.666**	**0.765**	**0.98**	-	-	-	-	-	-	-
P18	0.144	0.007	0.007	-	0.05	0.05	-	0.273	0.248	-	**0.771**	0.131	0.061	**0.746**	**0.843**	**0.89**	**0.907**	-	-	-	-	-	-
P19	0.126	0.006	0.005	-	0.121	0.121	-	0.234	0.194	-	**0.708**	0.123	0.078	**0.799**	**0.891**	**0.696**	**0.71**	**0.745**	-	-	-	-	-
P20	0.111	0.007	0.003	-	0.001	0.001	-	0.111	0.248	-	0.248	0.177	0.102	0.212	0.304	0.321	0.329	0.332	0.236	-	-	-	-
P21	0.028	0.001	0.001	-	0	0	-	0.032	0.012	-	0.088	0.139	**0.306 ***	0.103	0.105	0.079	0.08	0.088	0.028	0.194	-	-	-
P22	0.226	0.006	0.006	-	0.263	0.263	-	0.038	0.101	-	0.001	0.111	0.067	0.016	0.014	0.007	0.007	0.012	0.028	0.24	0.097	-	-
P23	-	-	-	-	-	-	-	-	-	-	-	-	-	-	-	-	-	-	-	-	-	-	-

P1: c.-193T > C; P2: c.-136C > T; P3: c.105C > G; P4: c.249+32G > C; P5: c.249+36G > C; P6: c.249+39T > G; P7: c.249+64C > A; P8: c.249+320G > C; P9: c.249+350_351delCA; P10: c.249+359G > A; P11: c.249+367G > A; P12: c.249+372C > T; P13: c.249+395delG; P14: c.249+398G > C; P15: c.249+399C > A; P16: c.249+400A > G; P17: c.249+401G > T; P18: c.249+402T > G; P19: c.249+405G > A; P20: c.249+455G > T; P21: c.249+472G > C; P22: c.361T > C; P23: c.396C > T. Bold text indicates strong LD with > 0.3 value. * indicate Hanwoo-specific strong LD scores.

**Table 4 genes-12-01662-t004:** Linkage disequilibrium (LD) scores among 23 polymorphisms of the bovine *IFITM3* gene in Holstein cattle.

	P1	P2	P3	P4	P5	P6	P7	P8	P9	P10	P11	P12	P13	P14	P15	P16	P17	P18	P19	P20	P21	P22	P23
P1	-	-	-	-	-	-	-	-	-	-	-	-	-	-	-	-	-	-	-	-	-	-	-
P2	-	-	-	-	-	-	-	-	-	-	-	-	-	-	-	-	-	-	-	-	-	-	-
P3	0.021	-	-	-	-	-	-	-	-	-	-	-	-	-	-	-	-	-	-	-	-	-	-
P4	0.01	-	**0.498 ***	-	-	-	-	-	-	-	-	-	-	-	-	-	-	-	-	-	-	-	-
P5	**0.897**	-	0.02	0.01	-	-	-	-	-	-	-	-	-	-	-	-	-	-	-	-	-	-	-
P6	**0.897**	-	0.02	0.01	**1**	-	-	-	-	-	-	-	-	-	-	-	-	-	-	-	-	-	-
P7	0.002	-	0	0	0.002	0.002	-	-	-	-	-	-	-	-	-	-	-	-	-	-	-	-	-
P8	0.01	-	0	0	0.012	0.012	0	-	-	-	-	-	-	-	-	-	-	-	-	-	-	-	-
P9	-	-	-	-	-	-	-	-	-	-	-	-	-	-	-	-	-	-	-	-	-	-	-
P10	0.04	-	0.002	0.001	0.044	0.044	0.021	0.01	-	-		-	-	-	-	-	-	-	-	-	-	-	-
P11	0.008	-	0.006	0.028	0.009	0.009	0.028	0.22	-	0.005	-	-	-	-	-	-	-	-	-	-	-	-	-
P12	-	-	-	-	-	-	-	-	-	-	-	-	-	-	-	-	-	-	-	-	-	-	-
P13	-	-	-	-	-	-	-	-	-	-	-	-	-	-	-	-	-	-	-	-	-	-	-
P14	0.018	-	0.002	0.001	0.021	0.021	0.001	0.175	-	0.011	**0.568**	-	-	-	-	-	-	-	-	-	-	-	-
P15	0.013	-	0.001	0.001	0.014	0.014	0.001	0.261	-	0.001	**0.709**	-	-	**0.693**	-	-	-	-	-	-	-	-	-
P16	0.021	-	0.001	0.001	0.022	0.022	0.001	**0.301**	-	0.002	**0.604**	-	-	**0.607**	**0.876**	-	-	-	-	-	-	-	-
P17	0.024	-	0.001	0.001	0.025	0.025	0.001	0.287	-	0	**0.571**	-	-	**0.635**	**0.834**	**0.867**	-	-	-	-	-	-	-
P18	0.03	-	0.001	0.001	0.032	0.032	0.001	0.25	-	0	**0.61**	-	-	**0.597**	**0.879**	**0.84**	**0.799**	-	-	-	-	-	-
P19	0.005	-	0.005	0.026	0.006	0.006	0.001	0.203	-	0.001	**0.726**	-	-	**0.572**	**0.656**	**0.558**	**0.526**	**0.562**	-	-	-	-	-
P20	-	-	-	-	-	-	-	-	-	-	-	-	-	-	-	-	-	-	-	-	-	-	-
P21	-	-	-	-	-	-	-	-	-	-	-	-	-	-	-	-	-	-	-	-	-	-	-
P22	0.065	-	0.001	0	0.063	0.063	0	0.001	-	0.014	0.01	-	-	0.012	0.009	0.008	0.008	0.009	0.011	-	-	-	-
P23	0.001	-	0	0	0.001	0.001	0	0.002	-	0.002	0.007	-	-	0.009	0.006	0.006	0.006	0.007	0.008	-	-	-	-

P1: c.-193T > C; P2: c.-136C > T; P3: c.105C > G; P4: c.249+32G > C; P5: c.249+36G > C; P6: c.249+39T > G; P7: c.249+64C > A; P8: c.249+320G > C; P9: c.249+350_351delCA; P10: c.249+359G > A; P11: c.249+367G > A; P12: c.249+372C > T; P13: c.249+395delG; P14: c.249+398G > C; P15: c.249+399C > A; P16: c.249+400A > G; P17: c.249+401G > T; P18: c.249+402T > G; P19: c.249+405G > A; P20: c.249+455G > T; P21: c.249+472G > C; P22: c.361T > C; P23: c.396C > T. Bold text indicates strong LD with > 0.3 value. * indicates Holstein cattle-specific strong LD scores.

**Table 5 genes-12-01662-t005:** Haplotype frequencies of bovine *IFITM3* gene polymorphisms in Hanwoo.

P1	P2	P3	P4	P5	P6	P7	P8	P9	P10	P11	P12	P13	P14	P15	P16	P17	P18	P19	P20	P21	P22	P23	Hanwoo (*n* = 216)
T	C	C	G	C	G	C	G	Wt	G	G	C	Wt	G	C	A	G	T	G	G	G	C	C	32 (0.148)
C	C	C	G	G	T	C	G	Wt	G	G	C	Wt	G	C	A	G	T	G	G	G	C	C	25 (0.116)
C	C	C	G	G	T	C	G	Del	G	A	C	Wt	C	A	G	T	G	A	G	G	T	C	17 (0.079)
C	C	C	G	G	T	C	G	Wt	G	G	C	Wt	G	C	A	G	T	G	T	G	T	C	14 (0.065)
C	C	C	G	G	T	C	G	Wt	G	A	C	Wt	C	A	G	T	G	A	G	G	T	C	10 (0.046)
C	C	C	G	G	T	C	C	Wt	G	A	C	Wt	C	A	G	T	G	A	G	G	C	C	9 (0.042)
C	C	C	G	G	T	C	C	Wt	G	G	C	Wt	G	C	A	G	T	G	T	G	T	C	8 (0.037)
C	C	C	G	G	T	C	G	Wt	G	G	T	Wt	G	C	A	G	T	G	T	C	T	C	8 (0.037)
C	C	C	G	C	G	C	C	Wt	G	A	C	Wt	C	A	G	T	G	A	G	G	C	C	7 (0.032)
C	C	C	G	C	G	C	G	Wt	G	G	C	Del	G	C	A	G	T	G	T	C	T	C	7 (0.032)
C	C	C	G	C	G	C	G	Wt	G	G	C	Wt	G	C	A	G	T	G	T	G	T	C	6 (0.028)
C	C	C	G	G	T	C	G	Del	G	A	T	Wt	C	A	G	T	G	A	G	G	T	C	5 (0.023)
Others *																							68 (0.315)

P1: c.-193T > C; P2: c.-136C > T; P3: c.105C > G; P4: c.249+32G > C; P5: c.249+36G > C; P6: c.249+39T > G; P7: c.249+64C > A; P8: c.249+320G > C; P9: c.249+350_351delCA; P10: c.249+359G > A; P11: c.249+367G > A; P12: c.249+372C > T; P13: c.249+395delG; P14: c.249+398G > C; P15: c.249+399C > A; P16: c.249+400A > G; P17: c.249+401G > T; P18: c.249+402T > G; P19: c.249+405G > A; P20: c.249+455G > T; P21: c.249+472G > C; P22: c.361T > C; P23: c.396C > T. * Others contain rare haplotypes with frequency < 0.02.

**Table 6 genes-12-01662-t006:** Haplotype frequencies of bovine *IFITM3* gene polymorphisms in Holstein cattle.

P1	P2	P3	P4	P5	P6	P7	P8	P9	P10	P11	P12	P13	P14	P15	P16	P17	P18	P19	P20	P21	P22	P23	Holstein (*n* = 226)
T	C	C	G	C	G	C	G	Wt	G	G	C	Wt	G	C	A	G	T	G	G	G	C	C	98 (0.434)
C	C	C	G	G	T	C	G	Wt	G	G	C	Wt	G	C	A	G	T	G	G	G	C	C	37 (0.164)
T	C	C	G	C	G	C	G	Wt	A	G	C	Wt	G	C	A	G	T	G	G	G	C	C	22 (0.097)
T	C	C	G	C	G	C	C	Wt	G	A	C	Wt	C	A	G	T	G	A	G	G	C	C	8 (0.035)
C	C	C	G	G	T	C	G	Wt	G	G	C	Wt	G	C	A	G	T	G	G	G	T	C	7 (0.031)
T	C	C	G	C	G	C	G	Wt	A	A	C	Wt	C	A	G	T	G	A	G	G	C	C	5 (0.022)
Others *																							49 (0.217)

P1: c.-193T > C; P2: c.-136C > T; P3: c.105C > G; P4: c.249+32G > C; P5: c.249+36G > C; P6: c.249+39T > G; P7: c.249+64C > A; P8: c.249+320G > C; P9: c.249+350_351delCA; P10: c.249+359G > A; P11: c.249+367G > A; P12: c.249+372C > T; P13: c.249+395delG; P14: c.249+398G > C; P15: c.249+399C > A; P16: c.249+400A > G; P17: c.249+401G > T; P18: c.249+402T > G; P19: c.249+405G > A; P20: c.249+455G > T; P21: c.249+472G > C; P22: c.361T > C; P23: c.396C > T. * Others contain rare haplotypes with frequency < 0.02.

**Table 7 genes-12-01662-t007:** In silico annotations of polymorphism of the bovine *IFITM3* gene.

Polymorphism	Methods	Score	Prediction
c.361T > C (F121L)	PolyPhen-2	0.001	Benign
	PANTHER	2	Probably benign
	PROVEAN	−0.357	Neutral

## Data Availability

Data are available on reasonable request. Requests may be made to bhjeong@jbnu.ac.kr.

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
