# Peer review of "Regulatory Single Nucleotide Polymorphism of the Bovine IFITM3 Gene Induces Differential Transcriptional Capacities of Hanwoo and Holstein Cattle"

_genes, 2021, doi:10.3390/genes12111662_

Round 1

Reviewer 1 Report

Introduction, paragraph 1: the name of the virus that causes COVID-19 is SARS-CoV-2.

Introduction: it would be useful to also mention the function of IFITM3 (with citations) in germline development (see https://www.sciencedirect.com/topics/biochemistry-genetics-and-molecular-biology/ifitm3), especially is this could be linked to the Nkx2-1 transcription factor binding site the authors identified, as Nkx2-1 has functions in organ development, as noted in the discussion.

Discussion, paragraph 1: the authors note that "recent studies have reported that the IFITM3 protein is also involved in not only immune-related functions..."; it would be helpful to identify some of these studies.

Author Response

  1. Introduction, paragraph 1: the name of the virus that causes COVID-19 is SARS-CoV-2.

Response: Thank for the reviewer’s good comment. As suggested by the reviewer, we have changed “COVID-19” to “SARS-CoV-2” in the Introduction section [Page 5, line 7].

  1. Introduction: it would be useful to also mention the function of IFITM3 (with citations) in germline development (see https://www.sciencedirect.com/topics/biochemistry-genetics-and-molecular-biology/ifitm3), especially is this could be linked to the Nkx2-1 transcription factor binding site the authors identified, as Nkx2-1 has functions in organ development, as noted in the discussion.

Response: Thank for the reviewer’s good comment. As suggested by the reviewer, we have added the references in the References section [Page 22, lines 7-9] and the sentences in the Discussion section [Page 15, lines 12-13]

  1. Discussion, paragraph 1: the authors note that "recent studies have reported that the IFITM3 protein is also involved in not only immune-related functions..."; it would be helpful to identify some of these studies.

Response: Thank for the reviewer’s good comment. As suggested by the reviewer, we have added the references in the References section [Page 22, lines 7-9; Page 23, lines 11-14].

Reviewer 2 Report

A good manuscript on the chracterization of the IFITM3 gene in two breeds of cattle which no doubt provides valuable information for further studies in other breeds for sustainable cattle production. I have made some comments in the edited text to help the authors improve on the quality of the manuscript.

Author Response

Reviewer 2

  1. A good manuscript on the characterization of the IFITM3 gene in two breeds of cattle which no doubt provides valuable information for further studies in other breeds for sustainable cattle production. I have made some comments in the edited text to help the authors improve on the quality of the manuscript.

Response: Thank for the reviewer’s good comment. As suggested by the reviewer, we have modified the manuscript.

Reviewer 3 Report

The article was well prepared and written. Congratulations on your research. Minor remarks in the attached manuscript.

Author Response

Reviewer 3

  1. The article was well prepared and written. Congratulations on your research. Minor remarks in the attached manuscript.

Response: Thank for the reviewer’s good comment. As suggested by the reviewer, we have modified the manuscript.
